# Spatiotemporal brain complexity quantifies consciousness outside of perturbation paradigms

**Martin Breyton[1]\*, Jan Fousek[2], Giovanni Rabuffo[3], Pierpaolo Sorrentino[3], Lionel Kusch[3], Marcello Massimini[4,5], Spase Petkoski[3], Viktor Jirsa[3]\***

[1]Aix-Marseille Université, APHM, Inserm, INS Institut de Neurosciences des Systèmes, Hôpital Sainte-Marguerite, Service de Pharmacologie Clinique et Pharmacosurveillance, Marseille, France; [2]Central European Institute of Technology (CEITEC), Masaryk University, Brno, Czech Republic; [3]Aix-Marseille Université, Inserm, Institut de Neurosciences des Systèmes, UMR 1106, Marseille, France; [4]Department of Biomedical and Clinical Sciences, University of Milan, Milan, Italy; [5]IRCCS Fondazione Don Carlo Gnocchi, Milan, Italy

**\*For correspondence:**
martin.breyton@univ-amu.fr (MB);
viktor.jirsa@univ-amu.fr (VJ)

## eLife Assessment

This **important** study examined the complexity of emergent dynamics of large-scale neural network models after perturbation (perturbational complexity index, PCI) and used it as a measurement of consciousness to account for previous recordings of humans at various anesthetized levels. The evidence supporting the conclusion is **convincing** and constitutes a unified framework for different observations related to consciousness. There are many fields that would be interested in this study, including cognitive neuroscience, psychology, complex systems, neural networks, and neural dynamics.

**Abstract** Signatures of consciousness are found in spectral and temporal properties of neuronal activity. Among these, spatiotemporal complexity after a perturbation has recently emerged as a robust metric to infer levels of consciousness. Perturbation paradigms remain, however, difficult to perform routinely. To discover alternative paradigms and metrics, we systematically explore brain stimulation and resting-state activity in a whole-brain model. We find that perturbational complexity only occurs when the brain model operates within a specific dynamical regime, in which spontaneous activity produces a large degree of functional network reorganizations referred to as being fluid. The regime of high brain fluidity is characterized by a small battery of metrics drawn from dynamical systems theory and predicts the impact of consciousness-altering drugs (Xenon, Propofol, and Ketamine). We validate the predictions in a cohort of 15 subjects at various stages of consciousness and demonstrate their agreement with previously reported perturbational complexity, but in a more accessible paradigm. Beyond the facilitation in clinical use, the metrics highlight complexity properties of brain dynamics in support of the emergence of consciousness.

## Introduction

Wakefulness in healthy subjects is linked to motor behavior and consciousness. Under some pathological or physiological conditions, behavioral response might break down, but consciousness doesn't necessarily vanish. Anesthetic drugs (***Figure 1a***) are used to medically induce reversible states of

**Figure 1.** Conceptual framework of the study. (**a**) Consciousness is a continuum and can be explored with drug-induced coma of various depths (Xenon, Propofol >Ketamine > Wakefulness). We hypothesize a correspondence between the variations in complexity found with PCI and the dynamics of spontaneous activity across the spectrum of consciousness. (**b**) We sketch various patterns of spatio-temporal activity reflecting changes in perturbational complexity from left to right. In (**c**), we show the conceptual shapes of corresponding manifolds of brain activity responsible for different sizes of the functional repertoire (number of wells) and associated with consciousness. (**d**) The brain is modeled as a network of neural masses coupled by an empirical connectome. This whole-brain model serves as a platform to simulate resting state activity (bottom left) and cortical stimulation (top left, example of firing rate time series with applied stimulus). Dynamical properties of the simulations are studied and compared with data features of human empirical recordings of spontaneous activity (bottom right, EEG during wakefulness and under three anesthetics) and stimulation (top right, TMS-EEG protocol performed in the same conditions).

unconsciousness of various depths, typically characterized by unresponsiveness. Common anesthetic drugs like Xenon or Propofol lead to an unresponsive and unconscious state resembling NREM sleep (**Murphy et al., 2011**). Others, such as Ketamine, can cause an unresponsive yet active state similar to REM sleep where conscious experience remains possible even without behavioral response (**Hobson, 2009**). These states of disconnected consciousness can arise following brain injury and pose challenges in clinical practice for consciousness assessment and prognosis evaluation (**Edlow et al., 2021**; **Giacino et al., 2018**). In an attempt to solve this problem, recent studies have developed a technique based on transcranial magnetic stimulation (TMS, **Casali et al., 2013**), in which the brain is stimulated by TMS pulses (**Hallett, 2007**) and evoked responses are recorded with simultaneous electroencephalography (EEG). It is shown that the evoked response can qualitatively differ depending on the patient's state (awake, REM, anesthesia, etc.), and is characterized by a simple scalar metric: the perturbational complexity index (PCI). PCI measures the complexity of the distributed spatiotemporal response in the brain, with values close to 1 when the complexity is high, and closer to 0 for simpler responses. Notably, increasing evidence suggests the PCI index as a reliable marker (**Figure 1b**) to differentiate between awake individuals and those in NREM sleep or an anesthetic-induced unconsciousness (**Casarotto et al., 2016**; **Sarasso et al., 2015**).

Dynamical systems theory is becoming increasingly popular in neuroscience (**Breakspear, 2017**; **Breakspear, 2007**) to study the evolution of spontaneous brain activity, that is *resting state*. Resting-state activity is highly organized and displays structured patterns of regional activations in large-scale recordings across imaging modalities. Some co-activation patterns are consistent across individuals and are known as functional brain networks (**Damoiseaux et al., 2006**). Over time, these activation patterns can emerge and reappear, suggesting a rich temporal structure with unique characteristics across tasks and individuals (**Peng et al., 2023**). Functional Connectivity (FC) measures capture the

topographic organization of these patterns (*van den Heuvel and Hulshoff Pol, 2010*) and are static views of the functional coordination between brain regions. Their temporal organization is estimated by dynamic Functional Connectivity (dFC), generally measured by correlating FC patterns at distinct time windows (*Hansen et al., 2015*; *Preti et al., 2017*). In this context, data complexity relates to the range of activity patterns seen over time (*Ghosh et al., 2008*; *Jirsa and Sheheitli, 2022*), defining a functional repertoire (*Figure 1c*). A diverse functional repertoire, supporting complex dynamics, is characterized by the exploration of numerous brain states, found in specific dynamical regimes (*Hansen et al., 2015*; *Golos et al., 2015*; *Sorrentino et al., 2021b*). Further evidence suggests that the brain is self-regulating around the transition between ordered and disordered phases (*Cocchi et al., 2017*; *O'Byrne and Jerbi, 2022*), where a high fluidity results in a rich temporal structure. In this study, we aim to examine the connection between perturbational complexity and fluid dynamics near this transition.

For that purpose, we use a large-scale brain network model – The Virtual Brain (TVB, *Sanz Leon et al., 2013*) – to simulate non-invasive brain stimulation and spontaneous activity (*Figure 1d*). Various dynamical regimes are explored by varying a global coupling parameter that scales the influence of the structural connections as compared to the local regional dynamics. Earlier studies indicated that a properly scaled global coupling parameter can produce phenomena of co-activations between brain regions, realistic FC patterns, and non-trivial dFC, as seen in fMRI experiments (*Rabuffo et al., 2021*; *Lavanga et al., 2022*). Here, we quantify the response to simulated TMS across dynamical regimes using the *simulation* PCI (*sPCI*) and connect it to dFC metrics. We hypothesize that simulations can only produce high *sPCI* and non-trivial dFC when the system is fine-tuned to a particular fluid regime. We then verify our findings by comparing empirical spontaneous and stimulation data of 15 subjects, each acquired during wakefulness and various states of consciousness (Propofol anesthesia ($N = 5$), Xenon anesthesia ($N = 5$), Ketamine anesthesia ($N = 5$)).

## Results

### Large-scale brain model and dynamical regimes

To explore the relationship between resting state dynamics and brain responsiveness, we used a large-scale brain network model (*Sanz-Leon et al., 2015*). This model consists of 84 cortical and subcortical regions, each of them being conceptualized by a neural mass model (NMM). The NMM is composed of a set of stochastic nonlinear differential equations, which describes the evolution of both the average firing rate $r(t)$ and the average membrane potential $V(t)$ of an ensemble of quadratic integrate and fire (QIF) neurons (*Montbrió et al., 2015*). The equations ascribe the dynamics of the node to two fixed points. One is an oscillatory, high-firing state, the *upstate* (inward spiral), and the other is a low-firing rate *downstate* (stable fixed point). Every node is influenced by a stochastic noise component, allowing it to transition between these two states. When embedded into a brain network, the state of each node is also influenced by the afferent input from its neighbors through the connectome (*Sporns, 2011*). The intensity of input any region receives is weighted by the size of the bundle of white fiber tracts, as estimated by diffusion-weighted imaging (DWI). For our study, due to the lack of subject-specific DWI data, we employed a generic connectome from a healthy subject obtained from the Human Connectome Project (HCP, *Van Essen et al., 2013*).

Examining the system at the network level, spontaneous activity can present a variety of dynamics depending on two global parameters: the global coupling $G$ (which scales the connectome) and noise intensity (white noise with zero mean and standard deviation $\sigma$). We highlight characteristic regimes (*Figure 2*), showing a carpet plot of firing rate activity and the profile of cascades over time. The profile of cascades was computed as the instantaneous deviation from baseline activity (see Methods). In the low coupling regime (*Figure 2a*), regions of the network are weakly connected and mostly behave independently, driven by noise. Hence, when the noise is low (*Figure 2a*, bottom), the activity is generally low and sparse, as most of the nodes are in the down state, with some rare and transient activations. As noise increases (*Figure 2a*, top), activations are more frequent, yet stochastic and non-coherent. In contrast, when global coupling is set to a high value (*Figure 2c*), brain regions are tightly connected, and the activity of each node is heavily entrained by the rest of the network. In this regime, most nodes tend to stay in the *upstate* due to high average input (proportional to $G$, since $I(t) = f(G, W)$). Only nodes that are weakly connected to the rest of the network will stay in the

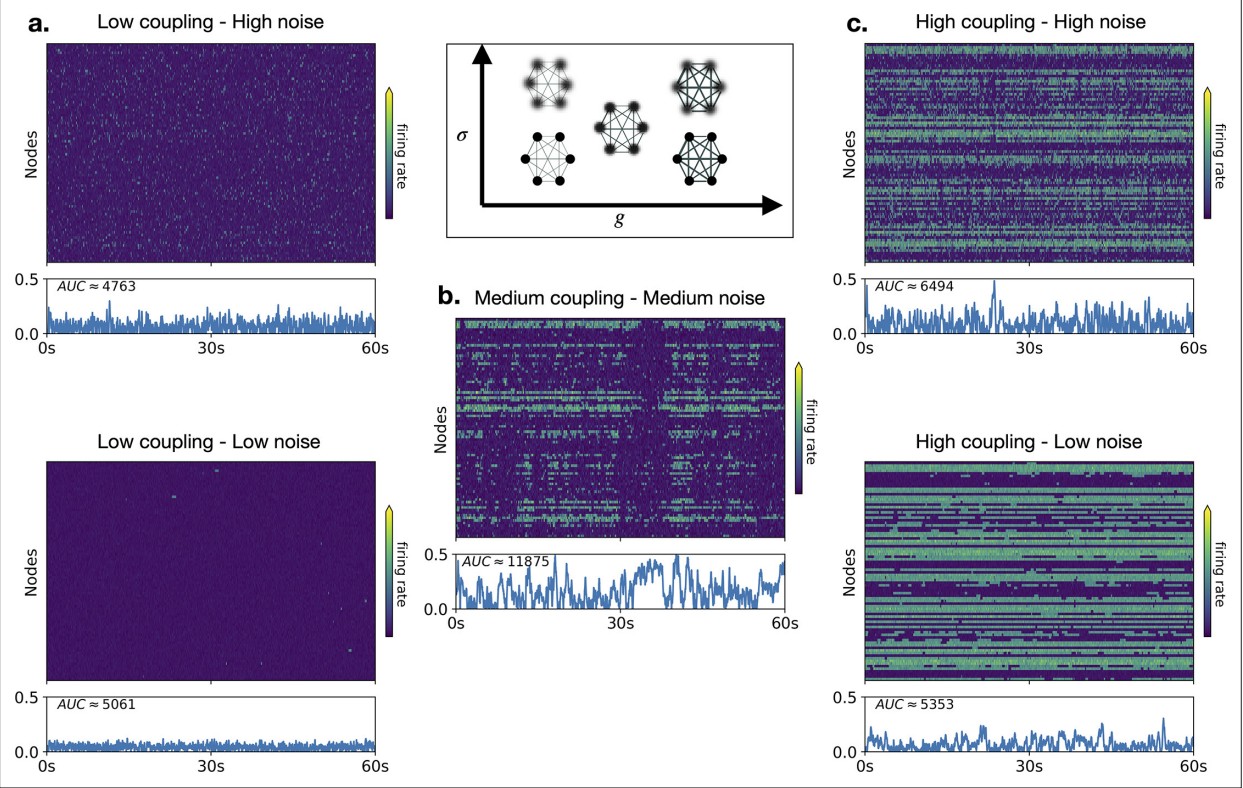

**Figure 2.** Dynamical regimes and cascades. Examples of firing rate time series of the model depending on the strength of interaction between nodes (global coupling, $G$) and noise intensity ($\sigma$). On the left column (**a**), the interactions are weak ($G = 0.27$, $\sigma = 0.022$ (bottom), $\sigma = 0.056$ (top)) and the activity is sparse. On the right column (**c**), connections are tighter (G=0.65, sigma-bottom=0.022, sigma-top=0.056) and a stable coactivation pattern appears in an ordered fashion. In the middle (**b**) (bottom), global coupling and noise are at optimal values (G=0.56, sigma = 0.036) and allow the emergence of structured patterns (coactivation cascades) of different sizes and durations. Below each time series, the blue line plot shows a quantification of cascades. It corresponds to the absolute value of the mean signal after z-scoring each node's activity. The AUC is the area under the blue line; it is a quantification of the presence of co-activation cascades.

The online version of this article includes the following figure supplement(s) for figure 2:

**Figure supplement 1.** Dynamical regimes and stimulation.

*downstate* despite high coupling. This behavior gives rise to a stable pattern of regional coactivations that involve characteristic subnetworks mirroring the structural connectivity (*Rabuffo et al., 2021*; *Pope et al., 2021*) and resulting in stereotyped activity on a slow timescale. However, in between these two regimes lies an intermediate regime, around a working point (*Figure 2b*), where cascades of coactivations of different sizes and durations occur in a complex and unpredictable fashion. Yet these coactivation patterns are not purely stochastic (*Battaglia et al., 2020*). They are often repetitive, revisiting the same neighborhood in state space (*Fousek et al., 2024*).

## Modeling stimulation and spontaneous activity

Brain stimulation was modeled by adding a transient external current to the membrane potential, mimicking a single pulse stimulus to a cortical brain region. When perturbed, the entire system shifts from its current state as the stimulus propagates through the network, until it settles into a new stable state. In line with the PCI experimental protocol, we sampled from multiple initial conditions and stimulated regions, presenting the maximum sPCI for each regime (i.e. each $\{G, \sigma\}$). For each simulation, we measured the complexity of the activity of the whole network over a 10 second period post-stimulus. We then compared this to the complexity of the baseline (i.e. the same simulation without a stimulus) using a ratio (see Methods). This helped us determine the relationship between the current dynamical regime and the system's ability to integrate a perturbation flexibly. We found that the maximum increase in complexity is reached when the global coupling is set around $G = 0.55$, and complexity itself can be increased fivefold with respect to baseline (*Figure 3*, top-left). If the system

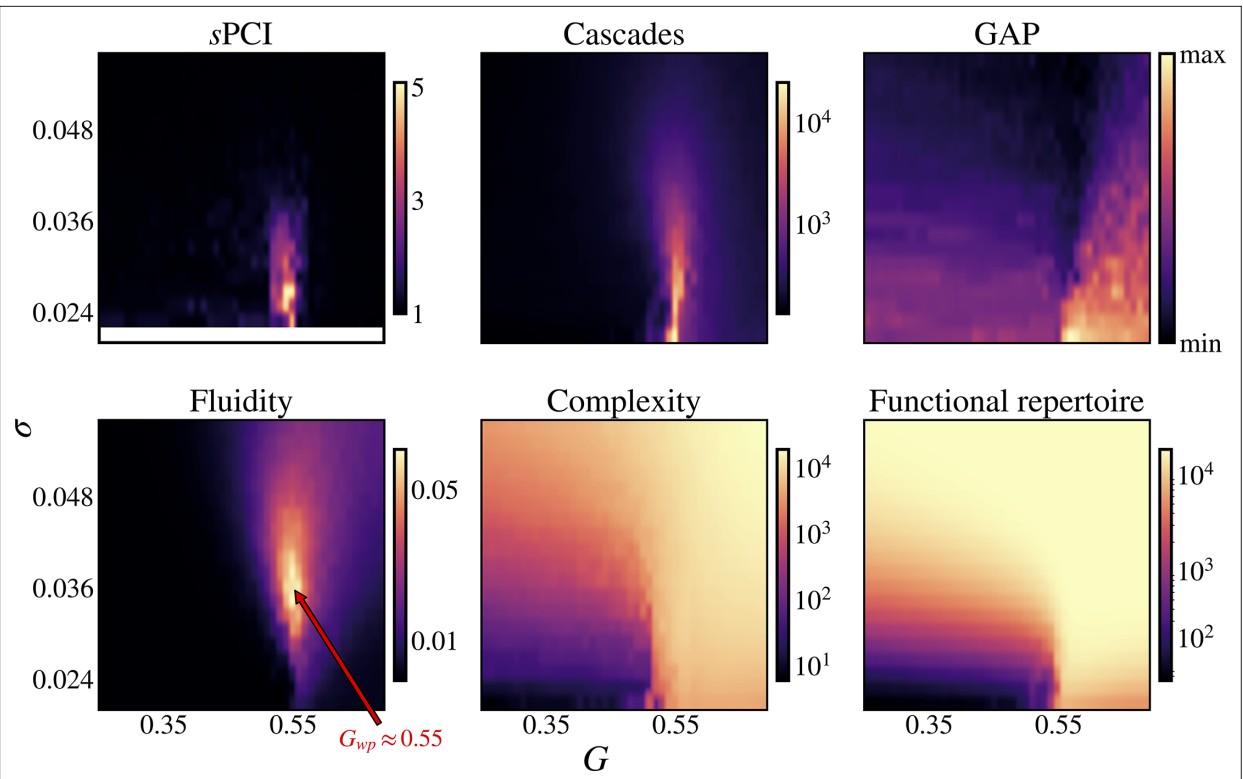

**Figure 3.** Exploration of stimulation and spontaneous activity in the parameter space. Metrics of spontaneous activity and following a perturbation in the parameter space of the model (G, sigma). Change of complexity after a stimulation measured by the sPCI (top left). Fluidity of the dynamics (bottom left) of spontaneous firing rate activity measured by $Variance(dFC)$ with a sliding window of $3s$ and $1s$ step size. The optimal point or working point where fluidity is maximal (bottom left, red arrow). The number of cascades (top middle) measured by the AUC of the z-scored absolute mean signal. Global Activation Potential (GAP; top-right) is assessed by the fastest change in the residual sum of squares across sources' membrane potential. Complexity of spontaneous activity (bottom middle) measured by the Lempel-Ziv complexity of binarized firing rate activity (with a threshold at $r = 0.7$). And the size of the functional repertoire (bottom right) defined by the number of unique patterns of binarized firing rate activity (with a threshold at $r = 0.7$).

operates far from this working point, the stimulus has minimal impact on global activity, meaning no notable change in complexity. The visual inspection of stimulation time series (1) across different regimes indicates that stimulation can trigger (or disrupt) coactivation cascades when the system is near maximum fluidity ($G = 0.55$). This behavior is responsible for a symmetry between minimum and maximum sPCI, as one can see that they share the same distribution in the parameter space. If sPCI is greater than one (lower than one), the trajectory of the system moves towards (away from) an attractor, thus triggering (destroying) a coactivation cascade. When sPCI is close to one, either stable attractors are absent, or it's challenging to deviate from the attractor, but the resulting response won't exceed the baseline activity's complexity. To compare responsiveness to resting state activity, we examined the fluidity of the system, defined by the variance of the upper triangular part of the dFC. Fluidity captures the recurring patterns of co-activation across time and, more specifically, the switches between highly correlated and non-correlated patterns. Even though direct evidence is sparse due to challenges in probing human brain activity's manifold, fluidity is believed to reflect the complexity of the energy landscape and the number of attractors (*Hansen et al., 2015*; *Fousek et al., 2024*). The distribution of fluidity in the model's parameter space (*Figure 3*, bottom-left) shows a peak near the working point value, reflecting recurring cascades during simulation, and decreases monotonously as the regime is drawn away from it.

To highlight that fluidity is a property of the dynamics and is not just tied to a single metric, we investigate and present three additional metrics characterizing spontaneous activity. We first calculate the raw Lempel-Ziv (LZ) complexity of each regime after binarization of the time series. LZ complexity counts the number of unique patterns (sequences of bits) found in a binary string, allowing for patterns of different sizes. A simple fixed threshold was used to binarize each time series into a sequence of

*upstates* and *downstates*. LZ complexity (*Figure 3*, bottom-middle) is found to increase abruptly as global coupling crosses the working point separating a low complexity-low coupling regime from a high complexity-high coupling regime. Second, we extracted the size of the functional repertoire of the model by counting the number of unique patterns that the network takes over time. As for LZ complexity, there is a separation of the space around the working point by a sharp increase of the functional repertoire (*Figure 3*, bottom-right), although this separation shifts towards lower coupling when noise increases and is lost for regimes with high noise. Lastly, we measured the Global Activation Potential (GAP) of the system. GAP aims at capturing the fastest change in global activity of the network, and it is computed as the maximum of the gradient of the residual sum of squares between sources (see Methods). Again, a similar pattern is seen with an increase around the working point (*Figure 3*, top-right). These tight relationships between simulation features (on spontaneous and stimulated activity) and the tuning of global coupling suggest that the same fluid dynamical regime underpins both optimal brain responsiveness and optimal spontaneous activity. Under the assumption that the global coupling models a free parameter in the brain, which is subject to change upon internal and/or external states, global dynamical changes should be found for different brain states. Based on this reasoning, we sought to retrace the signatures of these same dynamical regimes in real data in spontaneous EEG recordings during wakefulness and anesthesia.

## Data analysis

To do so, we used previously published experimental data, provided by *Sarasso et al., 2015*. 15 healthy subjects underwent a TMS-EEG protocol and a spontaneous EEG during anesthesia (Xenon, Propofol, Ketamine) and wakefulness. These anesthetics were posited as empirical models of loss of consciousness. Reports upon awakening were used as a ground truth for conscious experience, and it was found that subjects under Xenon or Propofol had no recollection of conscious experience, whereas Ketamine-induced anesthesia was always associated with reports of consciousness (i.e. dreaming). During the TMS-EEG protocol, multiple sessions of stimulation were performed, each consisting of multiple trials per stimulation site. A PCI value (across trials average) was obtained for each session, and the maximum PCI across sessions was retained. Only the maximum value of PCI for each subject and condition was available to us. A PCI at 0.31 was benchmarked as a threshold for consciousness (*Casarotto et al., 2016*).

As published previously, PCI successfully distinguishes conscious from unconscious states. Complexity measured with PCI is systematically higher during wakefulness than under anesthesia for Xenon and Propofol drugs, but not for Ketamine induced anesthesia, which is grouped with wakefulness ($PCI > 0.31$). Based on modeling results linking perturbational complexity and spontaneous activity metrics to the same underlying dynamical regimes modulated by global coupling, we hypothesized that variations of complexity should also be found at rest and associated with consciousness. We then studied the four resting-state metrics described previously to demonstrate that consciousness is associated with more complex activity at rest. For all subjects, at least one metric can perfectly distinguish wakefulness from anesthesia. We first extracted fluidity of the recordings on a broadband spectrum (high-pass filter at $0,5Hz$ and low-pass at $60Hz$). Dynamic functional connectivity was calculated using circular correlation between electrodes with a sliding window approach (1 s window, 100 ms step size; *Perdikis et al., 2025*). Then, for each subject and recording, we used a bootstrap procedure to sample uniformly with replacement 50 segments of 1 min. Complexity, the size of the functional repertoire, and GAP were also computed on each segment. When binarization was applied, we z-scored each electrode independently and thresholded at 3 standard deviations. This yielded a distribution of 50 points for each subject and condition (anesthesia or wakefulness). We present in *Figure 4* the results, subject-wise, for the four resting state metrics we studied: fluidity, complexity, functional repertoire, and GAP between anesthesia and wakefulness conditions. In *Figure 5*, we show classification results when grouping participants either on anesthesia (vs wakefulness) or conscious report (vs. no report). We cross-plot each metric against the PCI on the top and show classification accuracies of all metrics after fitting an SVM classifier with a linear kernel. For these last results, we computed complexity and the size of the functional repertoire on the full recordings and adjusted them by the length of the signal. According to our hypothesis and modeling results, we expected higher values of fluidity during wakefulness than anesthesia (*Figure 4a*). The separation between conditions was perfect for Propofol and Xenon anesthesia. Ketamine results were inconsistent, with subject 2 showing a reversed pattern

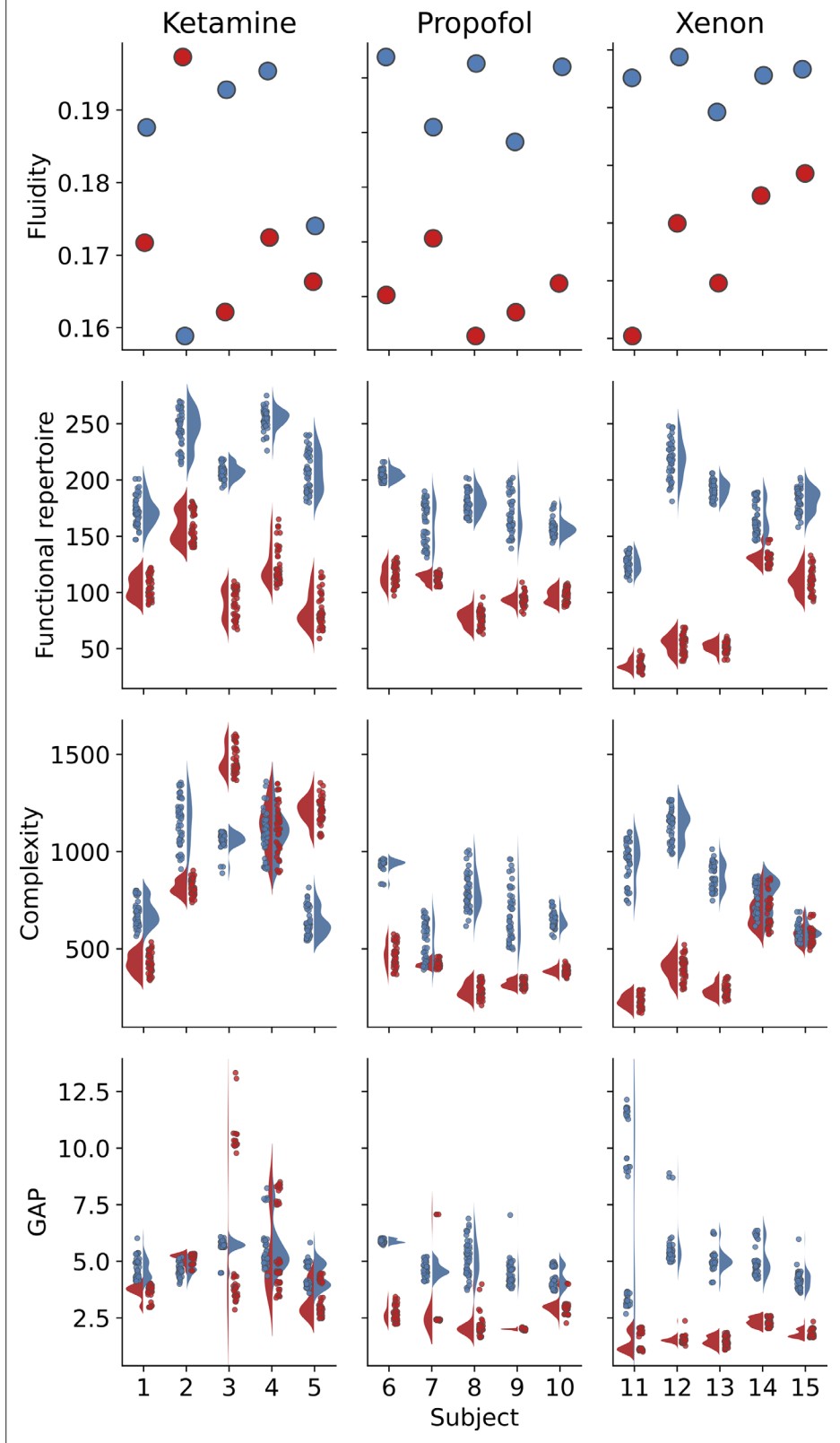

**Figure 4.** Resting-state metrics on EEG during anesthesia and wakefulness. Subject-specific distributions of
(**a**) fluidity, (**b**) the size of the functional repertoire, (**c**) Lempel-Ziv complexity, and (**d**) GAP, between wakefulness
(blue) and anesthesia (red). From left to right in each panel: Ketamine group, Propofol group, and Xenon group.
Values and distributions (kernel density estimations) were obtained by randomly sampling with replacement a

*Figure 4 continued on next page*

*Figure 4 continued*

minute of signal within each subject's recording (50 samples drawn per subject per condition, 1 point per sample). Fluidity was calculated on the full recordings for each subject.

(higher fluidity during anesthesia). Based on the results obtained on perturbational complexity, we expected to find systematically more spontaneous complexity during wakefulness than anesthesia (*Figure 4c*). For Propofol anesthesia, all subjects had a good separation, and for Xenon, three subjects were well classified, but two had overlapping distributions of complexity.

For Ketamine, subject 4 had overlapping distributions when the other 4 showed good separation, though the relationship was inverted for subjects 3 and 5, with less complexity found in wakefulness. In line with *Rabuffo et al., 2021*, we hypothesized that GAP was associated with the flexibility of the brain to switch between distant functional states and expected this ability to be lost during anesthesia. The results for GAP were satisfying for the Xenon and Propofol conditions but poor for all subjects under Ketamine (*Figure 4d*). As expected, when the separation occurs between wakefulness and anesthesia, the GAP is consistently higher during wakefulness. It is likely hinting at different mechanisms of action of the drugs on neural dynamics, in which the GAP is not altered by Ketamine. Lastly, as the size of the functional repertoire reflects the extent of explored functional patterns, we were expecting to find it associated with wakefulness, when the brain is more solicited and needs to reconfigure more often. Indeed (*Figure 4b*), it is the metric that permits a perfect separation between wakefulness and anesthesia for all subjects and conditions. The number of states explored during the recordings was consistently larger during wakefulness than anesthesia for all drugs and subjects. Classification results using a linear SVM are reported in *Figure 5*. We report the crossplots of PCI and each of the resting-state metrics for all subjects and conditions in *Figure 5a*. Each point corresponds to the

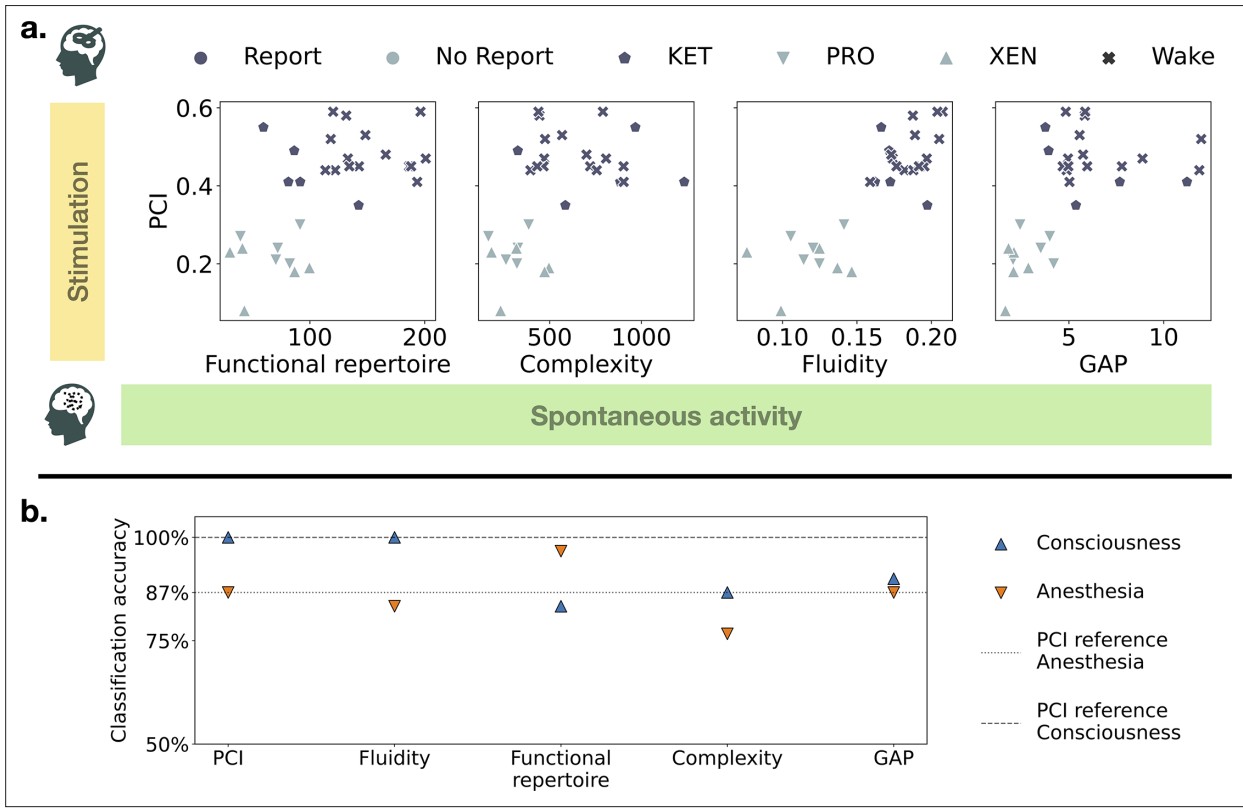

**Figure 5.** Predictive power of resting-state metrics and PCI. (**a**) Cross-plots between the PCI obtained experimentally during a TMS-EEG protocol and each metric on spontaneous recordings (functional repertoire, complexity, fluidity, and GAP). Complexity and the size of the functional repertoire were normalized by the length of the recording in minutes. (**b**) Classification accuracy of a Support Vector Machine classifier with a linear kernel to distinguish either between anesthesia and wakefulness (downward orange triangles) or between conscious report and no report (upward blue triangles). Dashed lines represent the benchmark performances achieved by PCI classification (100% for consciousness and 87% for anesthesia).

calculation of the given metric over the whole recording normalized by its duration. We find that for fluidity (*Figure 5a*, third panel), there is a perfect linear separation between Propofol and Xenon anesthesia on the left side and Wakefulness and Ketamine anesthesia on the right side. This corresponds to the classification accuracy result of 100% for the consciousness class in *Figure 5b*, which is the same for PCI. As expected, PCI and fluidity behave poorly at classifying the presence of an anesthetic agent due to the confusion induced by Ketamine. However, the size of the functional repertoire performs an almost perfect classification for this grouping. Only one subject under Ketamine has a high functional repertoire (*Figure 5a*, left panel), but all other subjects in the anesthesia condition have a size of functional repertoire roughly under 100. Classification accuracies for complexity and GAP at the group level are less performant but are shown for completeness.

## Discussion

Understanding the neural substrates of consciousness stands as one of the most significant scientific challenges encompassing both philosophical and theoretical dimensions. This challenge also stems from clinical practice, such as the need to characterize disorders of consciousness and their prognosis (*Edlow et al., 2021*; *Giacino et al., 2018*; *Sanders et al., 2012*). The detection of consciousness independently of sensory processing and motor behavior has recently improved with the development of the Perturbational Complexity Index that relies on the analysis of TMS-evoked potentials (*Casarotto et al., 2016*). Despite its remarkable performance, TMS-EEG can be challenging to implement in clinical practice as it requires complex equipment. In this work, we explored both the dynamics of spontaneous activity and responsiveness in a large-scale brain model and found strong evidence that common network-level factors govern the two. We demonstrate using a large-scale brain model that a similar dynamical regime of activity can cause perturbational complexity and complex (fluid) spontaneous activity. Using empirical data, we showed that spontaneous activity analyzed through appropriate metrics can be effective in assessing consciousness in wakefulness and anesthesia.

A large body of evidence highlights a relationship between the complexity of brain signals and conscious experience, as recently reviewed (*Sarasso et al., 2021*). The first theoretical argument started from a phenomenological description of conscious experience that led to theorizing the expected properties of the underlying neural activity (*Tononi and Edelman, 1998*; *Tononi et al., 1998*). In this view, the two building blocks of consciousness are functional integration (each experience is unified as a whole) and functional differentiation (each experience is unique and separated from others). These theoretical findings lead to the development of metrics rooted in information theory to estimate these functional properties on neural data. Of all these metrics, the PCI (relying on Lempel-Ziv complexity and entropy) revealed an interpretable and practical link between the spatio-temporal complexity of the propagation of a perturbation in the brain and the level of consciousness. In parallel, a large literature is dedicated to understanding brain function from the perspective of dynamical systems theory. In this field, the brain is viewed as a complex open system far from equilibrium composed of a large number of coupled components (e.g. neurons at the microscale or neural masses at the mesoscale) whose interactions are responsible for the emergence of macroscopic patterns. In such systems, drastic changes such as phase transitions or bifurcations can occur spontaneously or under the influence of a global parameter, with major effects on the dynamics. Brain simulations like the one presented in this work support the idea that a rich dynamics is possible when the system's parameters are fine-tuned around specific values (*Golos et al., 2015*). In fact, although debated, one prominent theory in neuroscience posits that the brain self-regulates near a critical regime, in the vicinity of a phase transition (*Cocchi et al., 2017*; *O'Byrne and Jerbi, 2022*). Simply put, this property allows the brain to be flexible enough to reconfigure and adapt dynamically to a changing environment, all while being stable enough to engage in complex sustained activities. Among other things, the critical regime is known to maximize information processing (*Beggs, 2008*) or to be related to cognitive processes (*Xu et al., 2022*), and criticality in the brain is also possibly linked to consciousness (*Tagliazucchi et al., 2016*; *Tagliazucchi, 2017*; *Toker et al., 2022*). Dynamical systems theory in neuroscience has also found a strong paradigm with the concept of manifolds. It's been shown that a high-dimensional nonlinear dynamical system displays low-dimensional but nevertheless complex behavior, which is equivalent to being constrained to an attractive low-dimensional hypersurface (aka manifold). Low-dimensional manifolds have been present in nonlinear dynamics for a long time, but as a concept for self-organization, it was first theorized in synergetics (*Haken,*

*1983*), but restricted to systems living close to a bifurcation; and later generalized to systems close to symmetries (*Pillai and Jirsa, 2017*; *Jirsa and Sheheitli, 2022*), notably symmetry breaking through large-scale connectivity in networks with time delays (*Petkoski et al., 2016*; *Petkoski et al., 2018*; *Petkoski and Jirsa, 2019*; *Petkoski and Jirsa, 2022*). Recent applications of this concept can be found in the domain of motor control (*Brennan and Proekt, 2019*) or in sleep/wake studies (*Chaudhuri et al., 2019*; *Rué-Queralt et al., 2021*). The link between dynamics and complexity is still under theoretical exploration (*Jirsa and Sheheitli, 2022*), and here we bring an empirical argument in that direction. Indeed, by using a large-scale brain model, we were able to control the dynamical regimes of the network and show that fluidity and responsiveness are maximal within the same parameter range. Fluidity captures the complexity of the manifold of the brain activity (i.e., number of attractors) and responsiveness quantifies the complexity upon perturbation. It could be argued that the noise level is not similar in both settings, but noise is only present to ensure continuous exploration of the manifold without affecting its topology (*Ghosh et al., 2008*). Complexity had already been shown to vary across conscious states (*Hudetz et al., 2016*), and the 1 /f slope of the EEG spectrum was also correlated with PCI on the same data we used (*Colombo et al., 2019*). Our validation results are in line with previous work on spontaneous activity and consciousness. In addition, we demonstrate that it's possible to distinguish ketamine-induced unresponsiveness from wakefulness at the individual level with a single metric.

Several studies have employed computational modeling approaches to investigate the differences in brain dynamics across states of consciousness. These studies present varying degrees of physiological detail and focus on complementary aspects of unconsciousness. They start from simple abstract models (Ising model) addressing, for example, the increased correlation between structural and functional connectivity in anesthesia (*Stramaglia et al., 2017*), or oscillator-based models (Hopf model) capturing a brain state-dependent response to simulated perturbation (*Deco et al., 2018*). More neurobiologically realistic models (Dynamic Mean Field) have also been used to combine multi-modal imaging data together with receptor density maps to address the macroscopic effects of general anesthesia and their relationship to spatially heterogeneous properties of the neuronal populations (*Luppi et al., 2022*). Similarly, using anatomically constrained parameters for brain regions has already been shown to increase the predictive value of brain network models (*Wang et al., 2019*; *Kong et al., 2021*). Furthermore, employing biophysically grounded mean-field and spiking neuron models (AdEx) allows addressing phenomena propagating in effect across multiple scales of description such as the molecular effects of anesthetics targeting specific receptor types (*Sacha et al., 2025*). Related work has shown that adaptation successfully reproduces dynamical regimes coherent with NREM and wakefulness (*Cattani et al., 2023*) with corresponding realistic PCI values (*Goldman et al., 2021*). Here, we do not address these biological questions but rather give a proof of concept that large-scale brain models can help understand the dynamics related to brain function. We used a model derived from QIF neurons (*Montbrió et al., 2015*) that lacks biological parameters such as ion concentration or synaptic adaptation. Nevertheless, we demonstrate that even the symmetry breaking caused by the connectome is sufficient for setting the global working point of the brain, which then links the brain's capacity for generating complex behavior in the different paradigms, that is, rest and stimulation.

One major limitation of our study lies in the algorithm we used to assess complexity in the model. The empirical PCI calculation is done on the average evoked response following stimulation that only lasts a few hundred milliseconds before returning to baseline activity. In the model, we spanned this calculation over ten seconds, thus capturing a slower dynamic than in real data. Nevertheless, we believe it doesn't affect our statement, and in theory, the neural mass model we used doesn't have a specific timescale. In future work, some caveats of our work could also be addressed and remedied. First, the imperfect separation of groups for some of the metrics could be improved by personalized brain modeling. The size of the functional repertoire, for instance, distinguishes wakefulness from anesthesia (including Ketamine) but lacks predictive power at the group level. This could be solved using a personalized model including structural information and parameter inference. Second and last, more realistic parameters could be included in the models, such as neuromodulatory pathways (*Taylor et al., 2022*; *Kringelbach et al., 2020*) to improve explanatory power.

## Methods

### Whole-brain model

The brain was modeled in the form of a network of interacting brain regions. The equation of each node of the network is given by the following differential equation (w.r.t. time):

$$\dot{\Psi} = \mathcal{F}(\Psi_i, \{k\}) + G * \sum_j W_{ij} S(\Psi_j(t - \tau_{ij})) + \xi_i(t) \tag{1}$$

The state of each region $i$ evolves in time following its own dynamics $\mathcal{F}$ and the inputs it receives from all other nodes of the network through the connectivity $W$. The global coupling parameter $G$ scales the connectivity of the network. The connectivity was derived from empirical data, here using a random healthy subject from the Human Connectome Project (HCP, *Van Essen et al., 2013*). The connectome was extracted through Diffusion Tensor Imaging (DTI) and captures the tracts' weights and lengths between regions of interest (ROI) according to the Desikan Kiliani Atlas (*Desikan et al., 2006*). In the end, 84 cortical and subcortical regions were connected through an undirected graph. The dynamics $\mathcal{F}$ of each region of interest (ROI) of the brain was modeled by a neural mass model developed by *Montbrió et al., 2015*. This model is a derivation of the mean firing rate $r$ and membrane potential $v$ of an all-to-all coupled network of quadratic integrate and fire (QIF) neurons. It is exact in the thermodynamic limit ($N \rightarrow \infty$, with $N$ the number of neurons) and is described by a set of two differential equations:

$$\dot{r} = \frac{\Delta}{\pi} + 2rv \tag{2}$$

$$\dot{v} = v^2 + \bar{\eta} + Jr + I(t) - (\pi r)^2 \tag{3}$$

$\eta$ is a neuronal excitability parameter that follows a Lorentzian distribution for which the mode $\bar{\eta}$ appears in the mean field equation and $\Delta$ is the scale (inter-quartile interval) of the same distribution. The parameter $J$ is the weight of synaptic integration of the QIF neurons. And $I(t)$ is an external current function. All neural masses were constructed using the same parameter values: $\bar{\eta} = -5.0$, $\Delta = 1.0$, and $J = 15.0$. Regions models were connected through their membrane potential by setting:

$$I(t) = G * \sum_{m \neq n} W_{nm} r_m(t - \tau_{nm}) \tag{4}$$

where $W_{nm}$ is the weight (white fiber tract width) and $\tau_{nm}$ is the delay along the tract connecting the two regions $n$ and $m$.

### Data

We retrieved data from studies *Sarasso et al., 2015*; *Colombo et al., 2019*, where 15 healthy subjects between 18 and 28 years old (5 males) were randomly assigned to one of three groups (5 subjects per group) to undergo anesthesia induced by either Xenon, Propofol, or Ketamine. For each subject, a 60-channel EEG recorded spontaneous activity of the brain followed by a TMS-EEG session for PCI assessment, once before drug administration (wakefulness) and once under anesthesia. All subjects were unresponsive during anesthesia as measured by the Ramsay (*Ramsay et al., 1974*) scale score of 6. Participants were asked to report any previous conscious experience upon awakening. Xenon and Propofol anesthesia led to no report of consciousness, while Ketamine allowed for retrospective conscious reports (vivid ketamine-dreams). We retrieved the spontaneous EEG recordings for every subject during wakefulness and during anesthesia, as well as the maximum PCI value from the TMS-EEG session and the absence/presence of conscious experience. For more details on the experimental setup and preprocessing steps, see *Sarasso et al., 2015*.

### Coactivation cascades

The profile of cascades over time was computed, first by z-scoring each source activity, and second by averaging the absolute value of the activity across all sources. The quantification of cascades was then obtained by calculating numerically the Area Under the Curve (AUC) of the profile of cascades.

## Complexity

For complexity assessment in spontaneous activity, we calculated the Lempel-Ziv complexity (*Lempel and Ziv, 1976*) denoted as $LZ_L$ on binarized activity. This algorithm, often used for data compression, scans through the sequence of binary elements and counts the number of unique patterns. For synthetic data, the bistable nature of the Montbrió (*Montbrió et al., 2015*) model allowed us to binarize firing rate time series (both spontaneous activity and after stimulation) by simple thresholding. The threshold ($r = 0.7$) was determined empirically by visual inspection with the intent to limit false positive and false negative rates of *upstate* activity. For empirical data, the binarization of spontaneous EEG is done after z-scoring each electrode and thresholding at 3 standard deviations.

## Perturbational complexity

To capture the complexity of the activity after stimulation, we computed an index strongly inspired by Perturbational Complexity Index (PCI). PCI relies on a complex statistical procedure to threshold the deterministic evoked brain response following the perturbation (*Casali et al., 2013*). This procedure is applied after source reconstruction of the EEG signal and yields a binary matrix of source activities over time. This binary matrix is then flattened and fed to the following formula:

$$\text{PCI}_L = \frac{\text{LZ}_L}{C_L}, \text{ with } C_L = \frac{L * H(L, p)}{log_2 L} \tag{5}$$

where $LZ_L$ is the Lempel-Ziv complexity of a series of length $L$ and $C_L$ is the asymptotic Lempel-Ziv complexity for a random series of the same length $L$, which itself is proportional to the source entropy that would have generated the data with the same average activity $p$. A high value of PCI corresponds to a complex propagation of the evoked signal in time and space across the brain. This procedure is repeated at several scalp locations, and the maximum PCI value obtained is retained. In this work, the empirical (maximum) PCI for every participant was provided by *Sarasso et al., 2015*.

In synthetic data, we adapted the procedure to extract the maximum synthetic PCI (sPCI). For each $\{G, \sigma\}$ couple, we stimulated a given node $node_i$ at a given time (or trial) $t$ to obtain a $sPCI_{stim}$ value over a window of 10 seconds of activity:

$$s\text{PCI}_{stim}(node_i, t, G, \sigma) = \frac{\text{LZ}_{stim}}{C_{stim}} \tag{6}$$

To distinguish the deterministic response of the system from the spontaneous activity, each of these $sPCI_{stim}$ value was matched with a baseline value:

$$s\text{PCI}_{baseline}(t, G, \sigma) = \frac{\text{LZ}_{baseline}}{C_{baseline}} \tag{7}$$

which corresponds to the complexity associated with the spontaneous activity of the realization of the exact same simulation (same parameters, same noise seed) but without the stimulus. The ratio of those two quantities was considered as the change in complexity due to the stimulus:

$$s\text{PCI}(node_i, t, G, \sigma) = \frac{s\text{PCI}_{stim}(node_i, t, G, \sigma)}{s\text{PCI}_{baseline}(t, G, \sigma)} \tag{8}$$

A value of $sPCI > 1$ (respectively $sPCI < 1$) indicates an increase (respectively a decrease) in complexity in response to the stimulus. 17 nodes were randomly selected (for computational limitations) among the 68 cortical nodes and 140 trials were done for each node near the working point $G \in [0.52, 0.57]$ and $\sigma \in [0.022, 0.039]$, and 6 trials elsewhere. We then selected the maximum $sPCI$ across time and nodes as the final $sPCI$:

$$s\text{PCI}(G, \sigma) = \max_{node, t} s\text{PCI}(node_i, t, G, \sigma) \tag{9}$$

## Fluidity

The extent of the repertoire of functional patterns explored spontaneously by a network is measured by fluidity defined as the variance of the upper triangle of the dynamic functional connectivity matrix (dFC). A sliding window (of length $\tau_{win}$) of functional connectivity (FC) is applied:

$$FC_{ij}(t) = FC[s_i(t'), s_j(t')], \text{ with } t' \in [t - \frac{\tau_{win}}{2}; t + \frac{\tau_{win}}{2}] \tag{10}$$

so that $FC(t) \in \mathbb{R}^{N \times N}$, for $N$ sources or electrodes in the network.

This yields a stream of FC matrices across the length of the signal $s(t)$. Dynamic functional connectivity is then computed as the correlation between the upper triangular part of those FC matrices to extract recurrences of functional patterns across time:

$$dFC(t_1, t_2) = corr[UpperTri(FC(t_1)), UpperTri(FC(t_2))] \tag{11}$$

so that $dFC(t) \in \mathbb{R}^{M \times M}$, for $M$ windows.

To finish, fluidity corresponds to the variance of the upper triangle of the dFC matrix, after removal of overlapping windows:

$$fluidity = Var[UpperTri(dFC - B]$$

$$B_{ij} = \begin{cases} 1 \text{ if } [t_i - \frac{\tau_{win}}{2}; t_i + \frac{\tau_{win}}{2}] \cap [t_j - \frac{\tau_{win}}{2}; t_j + \frac{\tau_{win}}{2}] \neq \emptyset \\ 0 \text{ otherwise} \end{cases}$$

In practice, the diagonal offset to compute fluidity is given by $offset = \frac{n_{overlap}}{\tau_{win} - n_{overlap}} + 1$, with $n_{overlap}$ the number of overlapping time points between two adjacent windows.

In synthetic data, functional connectivity was based on the zero-lag Pearson correlation between the firing rate activity of every pair of nodes of the network:

$$FC_{ij}(t) = corr[r_i(t'), r_j(t')], \text{ with } t' \in [t - \frac{\tau_{win}}{2}; t + \frac{\tau_{win}}{2}]. \tag{12}$$

In empirical EEG recordings, functional connectivity was assessed by the circular correlation coefficient among all pairs of electrodes ($N_{con} = 1770$ for 60 channels) over a sliding window of length $\tau_{win} = 0.55s$. We computed the Hilbert transform of each electrode to extract the instantaneous phase $\varphi(t)$ and mean phase within each window on a broadband spectrum $(0.5 - 60Hz)$. Then, between for each pair of electrodes $i, j$ we computed functional connectivity:

$$FC_{ij} = Ccor_{i,j}(t) = \frac{\sum_{t \in \tau_{win}} \sin(\varphi_i(t) - \bar{\varphi}_i) \sin(\varphi_j(t) - \bar{\varphi}_j)}{\sqrt{\sum_{t \in \tau_{win}} \sin^2(\varphi_i(t) - \bar{\varphi}_i) \sin^2(\varphi_j(t) - \bar{\varphi}_j)}} \tag{13}$$

Fluidity is related to previously defined metrics such as functional connectivity variability (*Müller et al., 2020*) that relied on a non-overlapping windowing procedure. We chose the term fluidity to convey a concept linked to dynamical systems in general and states exploration.

## Functional repertoire

To measure the size of the functional repertoire in empirical data, we first extracted the avalanches. The procedure is to z-score and threshold the EEG recording to obtain a binarized time series of activations (1 if $z(t) > 3 \, s.d$, 0 otherwise). An avalanche is a short-lived event that starts when an electrode is activated and ends when all electrodes return below the threshold. The method requires time to be binned and checking that the sequence of activation within an avalanche follows a branching process, by computing the branching ratio for each avalanche ($\sigma_i$) and averaging it (geometrically) across avalanches ($\sigma$):

$$\sigma_i = \prod_{j=1}^{N_{bin}-1} \left( \frac{n_{events}(j+1)}{n_{events}(j)} \right)^{\frac{1}{N_{bin}-1}}$$

$$\sigma = \prod_{i=1}^{N_{aval}} (\sigma_i)^{\frac{1}{N_{aval}}}$$

where $N_{bin}$ is the number of discrete time bins for the $i^{th}$ avalanche, $n_{event}$ the number of sources activated during a given time bin and $N_{aval}$ is the total number of avalanches in a recording. A branching ratio $\sigma = 1$ indicates a critical process and $\sigma < 1$ or $> 1$ a subcritical or supercritical process, respectively. We selected a bin size of 1 for which $\sigma = 1$ for all recordings. An avalanche pattern is defined as the set of regions that were recruited in a particular avalanche. The functional repertoire consists of all the distinct patterns (i.e., discarding repetitions) that occurred over time (*Sorrentino et al., 2021a*). In synthetic data, avalanche extraction was not always possible, depending on the dynamical regime. Hence, we simplified the procedure by counting the number of unique patterns across all time points after binarization.

## Global activation potential

Various modalities have shown that structured brain patterns emerge in short-lived bursts of neuronal activations (*Palva et al., 2013*; *Zamani Esfahlani et al., 2020*; *Beggs and Plenz, 2003*; *Tagliazucchi et al., 2012*). To capture these events in our data, we first z-score the data along time, and then perform the root-sum-square (RSS) across electrode:

$$RSS(t) = \sqrt{\sum_i z_i(t)^2} \tag{14}$$

where $z_i(t)$ is the z-scored activity. The RSS profile has values described by a fat-tail distribution and presents a bursting or activation profile characterized by rare sudden jumps. Similar to the PCI, which measures the potential for the system to express complex responses by selecting the maximum of perturbational complexity across an array of stimulations, here we define the potential of spontaneous activation for the system by selecting the maximum variation of the RSS, which we dub Global Activation Potential (GAP):

$$GAP = \max_t \left| \frac{d\,RSS(t)}{dt} \right| \tag{15}$$

## Classification

To test the classification accuracy of the resting state metrics we used a Support Vector machine algorithm with a linear kernel. Two different classifiers were evaluated. The first grouped all participants in the *wakefulness* condition (N=15) against the *anesthesia* condition (Ketamine N=5, Xenon N=5, and Propofol N=5). The second, grouping participants in *conscious report* (wakefulness N=15, and Ketamine anesthesia N=5) against *no report* (Xenon N=5, and Propofol N=5 anesthesia). Each classifier was trained using one feature at a time to find the optimal threshold for classification: PCI, fluidity, complexity, size of the functional repertoire, and GAP. All features were calculated on the full duration of each recording. Complexity and Functional repertoire features were normalized by the length in minutes.

## Acknowledgements

This research has received funding from EU's Horizon 2020 Framework Programme for Research and Innovation under the Specific Grant Agreements No. 101147319 (EBRAINS 2.0 Project), No. 101137289 (Virtual Brain Twin Project), No. 101057429 (project environMENTAL), and government grant managed by the Agence Nationale de la Recherche reference ANR-22-PESN-0012 (France 2030 program). M.M. is supported by the European Research Council (ERC-2022-SYG – 101071900 – NEMESIS) and by the Ministero dell'Istruzione, dell'Università e della Ricerca PNRR – EBRAINS-Italy.

# Additional information

## Competing interests
Marcello Massimini: is co-founder and shareholder of Intrinsic Powers, Inc, a spin-off of the University of Milan. The other authors declare that no competing interests exist.

## Funding

| Funder | Grant reference number | Author |
|---|---|---|
| European Commission | 10.3030/101147319 | Viktor Jirsa |
| European Commission | 10.3030/101137289 | Viktor Jirsa |
| European Commission | 10.3030/101057429 | Viktor Jirsa |
| Agence Nationale de la Recherche | 22-PESN-0012 | Viktor Jirsa |
| European Research Council | 10.3030/101071900 | Marcello Massimini |
| Ministero dell'università e della ricerca | EBRAINS-Italy | Marcello Massimini |

The funders had no role in study design, data collection and interpretation, or the decision to submit the work for publication.

## Author contributions
Martin Breyton, Conceptualization, Investigation, Visualization, Methodology, Writing – original draft, Writing – review and editing; Jan Fousek, Conceptualization, Supervision, Methodology, Writing – review and editing; Giovanni Rabuffo, Visualization, Methodology, Writing – review and editing; Pierpaolo Sorrentino, Lionel Kusch, Methodology; Marcello Massimini, Methodology, Writing – review and editing; Spase Petkoski, Conceptualization, Supervision, Writing – review and editing; Viktor Jirsa, Conceptualization, Supervision, Funding acquisition, Project administration, Writing – review and editing

## Author ORCIDs
Martin Breyton https://orcid.org/0000-0002-7979-3861
Jan Fousek https://orcid.org/0000-0002-8371-2956
Giovanni Rabuffo https://orcid.org/0000-0003-3947-1662
Pierpaolo Sorrentino https://orcid.org/0000-0002-9556-9800
Spase Petkoski https://orcid.org/0000-0003-4540-6293
Viktor Jirsa https://orcid.org/0000-0002-8251-8860

Reviewer #1 (Public review): https://doi.org/10.7554/eLife.98920.3.sa1
Reviewer #2 (Public review): https://doi.org/10.7554/eLife.98920.3.sa2
Author response https://doi.org/10.7554/eLife.98920.3.sa3

# Additional files

## Supplementary files
MDAR checklist

## Data availability
Raw EEG data used in this work is available on the repository Zenodo. The code to reproduce Figures 3 and 4 can be found on GitHub (copy archived at *Breyton, 2025*).

The following previously published dataset was used:

| Author(s) | Year | Dataset title | Dataset URL | Database and Identifier |
|-----------|------|---------------|-------------|------------------------|
| Massimini M, Laureys S | 2017 | Rest EEG recordings in healthy subjects during wakefulness, sleep and anesthesia with ketamine, propofol, and xenon | https://doi.org/10.5281/zenodo.806176 | Zenodo, 10.5281/zenodo.806176 |

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
