## [Editor Report · eLife Assessment]

This **important** study examined the complexity of emergent dynamics of large-scale neural network models after perturbation (perturbational complexity index, PCI) and used it as a measurement of consciousness to account for previous recordings of humans at various anesthetized levels. The evidence supporting the conclusion is **convincing** and constitutes a unified framework for different observations related to consciousness. There are many fields that would be interested in this study, including cognitive neuroscience, psychology, complex systems, neural networks, and neural dynamics.

---

## [Referee Report · Reviewer #1 (Public review)]

Summary:

This paper attempts to measure the complex changes of consciousness in the human brain as a whole. Inspired by the perturbational complexity index (PCI) from classic research, authors introduce simulation PCI (𝑠𝑃𝐶𝐼) of a time series of brain activity as a measure of consciousness. They first use large-scale brain network modeling to explore its relationship with the network coupling and input noise. Then the authors verify the measure with empirical data collected in previous research.

Strengths:

The conceptual idea of the work is novel. The authors measure the complexity of brain activity from the perspective of dynamical systems. They provide a comparison of the proposed measure with four other indexes. The text of this paper is very concise, supported by experimental data and theoretical model analysis.

Comments on revisions:

The manuscript is in good shape after revision. I would suggest that the author open-source the code and data in this study.

---

## [Referee Report · Reviewer #2 (Public review)]

Summary:

Breyton and colleagues analysed the emergent dynamics from a neural mass model, characterised the resultant complexity of the dynamics, and then related these signatures of complexity to datasets in which individuals had been anaesthetised with different pharmacological agents. The results provide a coherent explanation for observations associated with different time series metrics, and further help to reinforce the importance of modelling when integrating across scientific studies.

Strengths:

* The modelling approach was clear, well-reasoned and explicit, allowing for direct comparison to other work and potential elaboration in future studies through the augmentation with richer neurobiological detail.

* The results serve to provide a potential mechanistic basis for the observation that Perturbational Complexity Index changes as a function of consciousness state.

Weaknesses:

* Coactivation cascades were visually identified, rather than observed through an algorithmic lens. Given that there are numerous tools for quantifying the presence/absence of cascades from neuroimaging data, the authors may benefit from formalising this notion.

* It was difficult to tell, graphically, where the model's operating regime lay. Visual clarity here will greatly benefit the reader.

Comments on revisions:

The authors have addressed my concerns.

---

## [Author Response]

The following is the authors’ response to the original reviews

**Public Reviews:**

**Reviewer 1 (Public review):**
Summary:This paper attempts to measure the complex changes of consciousness in the human brain as a whole. Inspired by the perturbational complexity index (PCI) from classic research, authors introduce simulation PCI (_s_PCI) of a time series of brain activity as a measure of consciousness. They first use large-scale brain network modeling to explore its relationship with the network coupling and input noise. Then the authors verify the measure with empirical data collected in previous research.Strengths:The conceptual idea of the work is novel. The authors measure the complexity of brain activity from the perspective of dynamical systems. They provide a comparison of the proposed measure with four other indexes. The text of this paper is very concise, supported by experimental data and theoretical model analysis.

We would like to thank the reviewer for evaluation of our work and the positive feedback. In what follows we would like to clarify the ambiguities in our initial submission, and the respective changes to the manuscript.

(1) Consciousness is a network phenomenon. The measure defined by the authors is to consider the maximal sPCI across the nodes stimulated. This measure is based on the time series of one node. The measure may be less effective in quantifying the ill relationship between nodes. This may contribute to the less predictive power of anesthesia (Figure 4b).

Thank you for this comment, consciousness is indeed a network phenomenon. sPCI is in fact measured across the whole network: to compute sPCI we apply PCI to simulated activity of the whole network. The perturbation is applied to individual nodes of network (different node for each trial) and each time, the response to the stimulus is measured through sPCI in the whole network. To make this explicit, the relevant section now reads:

“In line with the PCI experimental protocol, we sampled from multiple initial conditions and stimulated regions, presenting the maximum sPCI for each regime (i.e., each {*G,σ*}). For each simulation, we measured the complexity of the activity of the whole network over a 10-second period post-stimulus.”

(2) One of the focuses of the work is the use of a dynamic model of brain networks. The explanation of the model needs to be in more detail.

Thank you for your feedback. We expanded the method section.

(3) The equations should be checked. For example, there should be no max on the left side of the first equation on page 13.

We thank the reviewer for spotting this typo, and we removed the max on the left side of this equation, and also checked all the other equations for correctness. The equation now reads:

\begin{document}$s \mathrm{PCl}(G, \sigma)=\max _{n o d e, t} s \mathrm{PCl}\left(\text {node}_{i}, t, G, \sigma\right)$\end{document}

(4) The quality of the figures should be improved.

Thank you for your comment. We have made adjustments to several figures and we hope that they are clearer now...

(5) Figure 4 should be discussed and analyzed more in the text.

Thank you for pointing this out. We added the following paragraph discussing the figure (now number 5) in the results section:

“Classification results using a linear SVM are reported in Fig. 5. We report the crossplots of PCI and each of the resting-state metrics for all subjects and conditions in Fig. 5a. Each point corresponds to the calculation of the given metric over the whole recording normalized by its duration. We find that for fluidity (Fig. 5a, third panel), there is a perfect linear separation between Propofol and Xenon anesthesia on the left side and Wakefulness and Ketamine anesthesia on the right side. This corresponds to the classification accuracy result of 100% for the consciousness class in Fig. 5b, which is the same for PCI. As expected, PCI and fluidity behave poorly at classifying the presence of an anesthetic agent due to the confusion induced by Ketamine. However, the size of the functional repertoire performs an almost perfect classification for this grouping. Only one subject under Ketamine has a high functional repertoire (Fig. 5a, left panel), but all other subjects in the anesthesia condition have a size of functional repertoire roughly under 100. Classification accuracies for complexity and GAP at the group level are less performant but are shown for completeness.”

(6) The usage of the terms PCI and sPCI should be distinguished.

We would like to thank the reviewer for pointing out this ambiguity. The PCI metric had to be adapted for the synthetic data. We have now further emphasized this in the methods sections – “Perturbational Complexity”.

**Reviewer 2 (Public review):**
Summary:Breyton and colleagues analysed the emergent dynamics from a neural mass model, characterised the resultant complexity of the dynamics, and then related these signatures of complexity to datasets in which individuals had been anaesthetised with different pharmacological agents. The results provide a coherent explanation for observations associated with different time series metrics, and further help to reinforce the importance of modelling when integrating across scientific studies.Strengths:(1) The modelling approach was clear, well-reasoned, and explicit, allowing for direct comparison to other work and potential elaboration in future studies through the augmentation with richer neurobiological detail.(2) The results serve to provide a potential mechanistic basis for the observation that the Perturbational Complexity Index changes as a function of the consciousness state.

We would like to thank the reviewer for assessing our work, and the valuable feedback.

Weaknesses:

(3) Coactivation cascades were visually identified, rather than observed through an algorithmic lens. Given that there are numerous tools for quantifying the presence/absence of cascades from neuroimaging data, the authors may benefit from formalising this notion.

Thank you for bringing this to our attention. We added a quantification of the cascades in Fig 2 and 3. We computed the absolute value of the mean signal across sources (following z-scoring) to obtain a cascade profile and calculated the area under the curve as quantification of the overall presence of cascades. As it can be seen in the two figures, the presence of cascades is the highest around the working point. We have also added the precise definition to the methods section, which now reads:

“Coactivation Cascades

The profile of cascades over time was computed, first by z-scoring each source activity, and second by averaging the absolute value of the activity across all sources. The quantification of cascades was then obtained by calculating numerically the Area Under the Curve (AUC) of the profile of cascades.”

(4) It was difficult to tell, graphically, where the model’s operating regime lay. Visual clarity here will greatly benefit the reader.

Thank you for pointing out this ambiguity, we have marked the working point explicitly in the Figure 3.

**Recommendations For The Authors**

**Reviewer 1 (Recommendations for the authors):**
(1) In the method section, the technical details of the other four indexes should be elaborated.

Thank you for your recommendation, we agree that the description in the submitted manuscript was too brief. We expanded the method section about the functional repertoire and the bursting potential.

**Reviewer 2 (Recommendations for the authors):**
(1) The authors could more clearly label the ”working point” of their parameter space. Perhaps a label/arrow on Figure 2c that directs the readers’ eyes towards the location in state-space that you define as the working point?

Thank you for pointing out this ambiguity, we updated the figure 3 to mark the working point precisely.

(2) While ’fluidity’ is quite an evocative term and does a great job of suggesting to the uninitiated reader the character of the time series in question, I wonder whether a more descriptive term might be better suited for this variable, even if as an adjunct to the term, fluidity. In the past, we (and others) have used the term dynamic functional connectivity variability (Mu¨ller et al., 2020 NeuroImage) to refer to this feature, as it links the measure directly to the technique from which it was estimated.

Thank you for your feedback. You are correct, dynamic functional connectivity variability could have been a wording of choice for some of our results. However the term “fluidity” was chosen to convey a broader theoretical concept linked to dynamical systems but not exclusive to the brain. Here, dynamic functional connectivity variability is merely a measure of the fluidity of the system. We added the following in the method section describing the metrics:

“[...] Fluidity is related to previously defined metrics such as functional connectivity variability [10] that relied on a non-overlapping windowing procedure. We chose the term fluidity to convey a cocept linked to dynamical systems in general and states exploration. [...]”

(3) The term ”bursting potential” is also potentially problematic, as ”bursting” refers to a different concept at the cellular level (i.e., multiple action potentials in a short window of time) than it does in the context that the authors are presumably using it here (i.e., the capacity for the dynamics of the population to ”burst” into the fat-tail of their activity distribution). To avoid ambiguity, it could be worth considering altering this terminology, perhaps again by using a term that is descriptive of the technique used to estimate it, rather than the concept that it evokes.

Thank you for pointing out this ambiguity in the naming of the bursting potential. We have renamed it to “Global Activation Potential (GAP)” as we believe this term is a better description of the metric. We have switched to this term across the whole manuscript.

(4) There is a range of other modelling studies that have compared brain dynamics in the awake vs. anaesthetised patient. In my opinion, the reader would benefit from the ability to place this work into the broader context created by the literature, particularly as there are subtle (yet potentially important) differences in the models used in each case. Note - as this is a subjective opinion, I don’t view this as a crucial addition to the paper’s potential strength of evidence, though I do believe that it would have a positive effect on its potential impact.

We thank you for the suggestion. We have modified the before-to-last paragraph of the discussion to bring more context from the literature models of anethesia and wakefulness:

“Several studies have employed computational modeling approaches to investigate the differences in brain dynamics across states of consciousness. These studies present varying degrees of physiological detail and focus on complementary aspects of unconsciousness. They start from simple abstract models (Ising model) addressing for example the increased correlation between stuctural and functional connectivity in aneshesia [15], or oscillator-based models (Hopf model) capturing a brain state dependent response to simulated perturbation [4]. More neurobiologically realistic models (Dynamic Mean Field) have also been used to combine multimodal imaging data together with receptor density maps to address the macroscopic effects of general aneshesia and their relationship to spatially heterogeneous properties of the neuronal populations [8]. Similarly, using anatomically constrained parameters for brain regions has already been shown to increase the predictive value of brain network models [6, 18]. Furthermore, employing biophysically grounded mean-field and spiking neuron models (AdEx) allows addressing phenomena propagating in effect across multiple scales of description such as the molecular effects of anesthetics targeting specific receptor types [12]. Related work has shown that adaptation successfully reproduces dynamical regimes coherent with NREM and wakefulness [3] with corresponding realistic PCI values Goldman2021comprehensive. Here, we don’t address these biological questions but rather give a proof of concept that large-scale brain models can help understand the dynamics related to brain function. We used a model derived from QIF neurons Montbrio2015Macroscopic that lacks biological parameters such as ion concentration or synaptic adaptation. Nevertheless, we demonstrate that even the symmetry breaking caused by the connectome is sufficient for setting the global working point of the brain, which then links the brain’s capacity for generating complex behavior in the different paradigms, that is, rest and stimulation.”

(5) I saw the label ”digital brain twin” in the abstract but then did not find a location in the main text/methods wherein this aspect of the modelling was explained.

Thank you for pointing out this discrepancy, we have removed the term “digital brain twin” and replaced it by “whole-brain model” everywhere.

References

John M. Beggs and Dietmar Plenz. Neuronal Avalanches in Neocortical Circuits. The Journal of Neuroscience, 23(35):11167–11177, dec 3 2003.

A. G. Casali, O. Gosseries, M. Rosanova, M. Boly, S. Sarasso, K. R. Casali, S. Casarotto, M.-A. Bruno, S. Laureys, G. Tononi, and M. Massimini. A Theoretically Based Index of Consciousness Independent of Sensory Processing and Behavior. Science Translational Medicine, 5(198):198ra105–198ra105, aug 14 2013.

Anna Cattani, Andrea Galluzzi, Matteo Fecchio, Andrea Pigorini, Maurizio Mattia, and Marcello Massimini. Adaptation shapes local cortical reactivity: From bifurcation diagram and simulations to human physiological and pathological responses. eneuro, 10(7):ENEURO.0435– 22.2023, July 2023.

Gustavo Deco, Joana Cabral, Victor M. Saenger, Melanie Boly, Enzo Tagliazucchi, Helmut Laufs, Eus Van Someren, Beatrice Jobst, Angus Stevner, and Morten L. Kringelbach. Perturbation of whole-brain dynamics in silico reveals mechanistic differences between brain states. NeuroImage, 169:46–56, April 2018.

Rahul S. Desikan, Florent S´egonne, Bruce Fischl, Brian T. Quinn, Bradford C. Dickerson, Deborah Blacker, Randy L. Buckner, Anders M. Dale, R. Paul Maguire, Bradley T. Hyman, Marilyn S. Albert, and Ronald J. Killiany. An automated labeling system for subdividing the human cerebral cortex on MRI scans into gyral based regions of interest. NeuroImage, 31(3):968–980, 7 2006.

Xiaolu Kong, Ru Kong, Csaba Orban, Peng Wang, Shaoshi Zhang, Kevin Anderson, Avram Holmes, John D. Murray, Gustavo Deco, Martijn van den Heuvel, and B. T. Thomas Yeo. Sensory-motor cortices shape functional connectivity dynamics in the human brain. Nature Communications, 12(1), November 2021.

A. Lempel and J. Ziv. On the Complexity of Finite Sequences. IEEE Transactions on Information Theory, 22(1):75–81, 1 1976. event-title: IEEE Transactions on Information Theory.

Andrea I. Luppi, Pedro A. M. Mediano, Fernando E. Rosas, Judith Allanson, John D. Pickard, Guy B. Williams, Michael M. Craig, Paola Finoia, Alexander R. D. Peattie, Peter Coppola, Adrian M. Owen, Lorina Naci, David K. Menon, Daniel Bor, and Emmanuel A. Stamatakis. Whole-brain modelling identifies distinct but convergent paths to unconsciousness in anaesthesia and disorders of consciousness. Communications Biology, 5(1), April 2022.

Ernest Montbri´o, Diego Paz´o, and Alex Roxin. Macroscopic Description for Networks of Spiking Neurons. Physical Review X, 5(2):021028, jun 19 2015.

Eli J. Mu¨ller, Brandon Munn, Luke J. Hearne, Jared B. Smith, Ben Fulcher, Aurina Arnatkeviˇciu¯te˙, Daniel J. Lurie, Luca Cocchi, and James M. Shine. Core and matrix thalamic sub-populations relate to spatio-temporal cortical connectivity gradients. NeuroImage, 222:117224, November 2020.

J. Matias Palva, Alexander Zhigalov, Jonni Hirvonen, Onerva Korhonen, Klaus LinkenkaerHansen, and Satu Palva. Neuronal long-range temporal correlations and avalanche dynamics are correlated with behavioral scaling laws. Proceedings of the National Academy of Sciences, 110(9):3585–3590, feb 26 2013. publisher: Proceedings of the National Academy of Sciences.

Maria Sacha, Federico Tesler, Rodrigo Cofre, and Alain Destexhe. A computational approach to evaluate how molecular mechanisms impact large-scale brain activity. Nature Computational Science, 5(5):405–417, May 2025.

Simone Sarasso, Melanie Boly, Martino Napolitani, Olivia Gosseries, Vanessa Charland-Verville, Silvia Casarotto, Mario Rosanova, Adenauer Girardi Casali, Jean-Francois Brichant, Pierre Boveroux, Steffen Rex, Giulio Tononi, Steven Laureys, and Marcello Massimini. Consciousness and Complexity during Unresponsiveness Induced by Propofol, Xenon, and Ketamine. Current Biology, 25(23):3099–3105, 12 2015.

Pierpaolo Sorrentino, Rosaria Rucco, Fabio Baselice, Rosa De Micco, Alessandro Tessitore, Arjan Hillebrand, Laura Mandolesi, Michael Breakspear, Leonardo L. Gollo, and Giuseppe Sorrentino. Flexible brain dynamics underpins complex behaviours as observed in Parkinson’s disease. Scientific Reports, 11(1):4051, feb 18 2021. number: 1 publisher: Nature Publishing Group.

S. Stramaglia, M. Pellicoro, L. Angelini, E. Amico, H. Aerts, J. M. Cort´es, S. Laureys, and D. Marinazzo. Ising model with conserved magnetization on the human connectome: Implications on the relation structure-function in wakefulness and anesthesia. Chaos: An Interdisciplinary Journal of Nonlinear Science, 27(4), April 2017.

Enzo Tagliazucchi, Pablo Balenzuela, Daniel Fraiman, and Dante Chialvo. Criticality in LargeScale Brain fMRI Dynamics Unveiled by a Novel Point Process Analysis. Frontiers in Physiology, 3, 2012. [Online; accessed 2022-12-23].

David C. Van Essen, Stephen M. Smith, Deanna M. Barch, Timothy E.J. Behrens, Essa Yacoub, and Kamil Ugurbil. The WU-Minn Human Connectome Project: An Overview. NeuroImage, 80:62–79, oct 15 2013. PMID: 23684880 PMCID: PMC3724347.

Peng Wang, Ru Kong, Xiaolu Kong, Rapha¨el Li´egeois, Csaba Orban, Gustavo Deco, Martijn P. van den Heuvel, and B.T. Thomas Yeo. Inversion of a large-scale circuit model reveals a cortical hierarchy in the dynamic resting human brain. Science Advances, 5(1), January 2019.

Farnaz Zamani Esfahlani, Youngheun Jo, Joshua Faskowitz, Lisa Byrge, Daniel P. Kennedy, Olaf Sporns, and Richard F. Betzel. High-amplitude cofluctuations in cortical activity drive functional connectivity. Proceedings of the National Academy of Sciences of the United States of America, 117(45):28393–28401, November 2020.